# Identification of a Novel Tumor Microenvironment Prognostic Signature for Advanced-Stage Serous Ovarian Cancer

**DOI:** 10.3390/cancers13133343

**Published:** 2021-07-03

**Authors:** Mingjun Zheng, Junyu Long, Anca Chelariu-Raicu, Heather Mullikin, Theresa Vilsmaier, Aurelia Vattai, Helene Hildegard Heidegger, Falk Batz, Simon Keckstein, Udo Jeschke, Fabian Trillsch, Sven Mahner, Till Kaltofen

**Affiliations:** 1Department of Obstetrics and Gynecology, University Hospital, LMU Munich, Marchioninistrasse 15, 81377 Munich, Germany; Mingjun.Zheng@med.uni-muenchen.de (M.Z.); Anca.ChelariuRaicu@med.uni-muenchen.de (A.C.-R.); Heather.Mullikin@ukr.de (H.M.); Theresa.Vilsmaier@med.uni-muenchen.de (T.V.); Aurelia.Vattai@med.uni-muenchen.de (A.V.); Helene.Heidegger@med.uni-muenchen.de (H.H.H.); Falk.Batz@med.uni-muenchen.de (F.B.); Simon.Keckstein@med.uni-muenchen.de (S.K.); Fabian.Trillsch@med.uni-muenchen.de (F.T.); Sven.Mahner@med.uni-muenchen.de (S.M.); 2Department of Liver Surgery, Peking Union Medical College Hospital, Chinese Academy of Medical Sciences & Peking Union Medical College, 9 Dongdan 3rd Alley, Dongcheng District, Beijing 100730, China; Lancet_Junyu@163.com; 3Department of Gynecology and Obstetrics, University Medical Center Regensburg, Landshuter Strasse 65, 93053 Regensburg, Germany; 4Department of Obstetrics and Gynecology, University Hospital Augsburg, Stenglinstrasse 2, 86156 Augsburg, Germany

**Keywords:** ovarian neoplasms, tumor microenvironment, genes, The Cancer Genome Atlas (TCGA), immunotherapy

## Abstract

**Simple Summary:**

The expression of tumor microenvironment-related genes is known to be correlated with ovarian cancer patients’ prognosis. Immunotherapeutic targets are in part located in this complex cluster of cells and soluble factors. In our study, we constructed a prognostic 11-gene signature for advanced serous ovarian cancer from tumor microenvironment-related genes through lasso regression. The established risk score can quantify the prognosis of ovarian cancer patients more accurately and is able to predict the putative biological response of cancer samples to a programmed death ligand 1 blocking immunotherapy. This might empower the role of immunotherapy in ovarian cancer through its usage in future study protocols.

**Abstract:**

(1) Background: The tumor microenvironment is involved in the growth and proliferation of malignant tumors and in the process of resistance towards systemic and targeted therapies. A correlation between the gene expression profile of the tumor microenvironment and the prognosis of ovarian cancer patients is already known. (2) Methods: Based on data from The Cancer Genome Atlas (379 RNA sequencing samples), we constructed a prognostic 11-gene signature (*SNRPA1*, *CCL19*, *CXCL11*, *CDC5L*, *APCDD1*, *LPAR2*, *PI3*, *PLEKHF1*, *CCDC80*, *CPXM1* and *CTAG2*) for Fédération Internationale de Gynécologie et d’Obstétrique stage III and IV serous ovarian cancer through lasso regression. (3) Results: The established risk score was able to predict the 1-, 3- and 5-year prognoses more accurately than previously known models. (4) Conclusions: We were able to confirm the predictive power of this model when we applied it to cervical and urothelial cancer, supporting its pan-cancer usability. We found that immune checkpoint genes correlate negatively with a higher risk score. Based on this information, we used our risk score to predict the biological response of cancer samples to an anti-programmed death ligand 1 immunotherapy, which could be useful for future clinical studies on immunotherapy in ovarian cancer.

## 1. Introduction

Ovarian cancer (OC) has the highest mortality rate among malignancies originating in the female reproductive system. Given the continuous improvement of both diagnostic techniques and treatment methods, the prevalence and survival rate of OC patients have increased during the past 15 years. More specifically, the prevalence of OC was recently reported to be 10 times higher than its incidence [1]. OC accounts for 3.3% of all malignant diseases in women in Germany and the cancer-related mortality rate is high, accounting for 5.6% of all malignancy-related deaths [2]. Advanced-stage serous ovarian cancer (SOC) is the cause of most OC deaths. Although 80% of patients significantly benefit from surgery and chemotherapy, they usually relapse and metastasize and are, therefore, incurable. Approximately two-thirds will experience a relapse after receiving several lines of therapy and die within 10 years of the first diagnosis [3,4].

At present, Fédération Internationale de Gynécologie et d’Obstétrique (FIGO), TNM stage and *BRCA1/2* status are the most relevant clinical variables to guide the management of SOC patients [5]. However, due to the high heterogeneity of SOC, which is the most important and representative entity, among others, the prognosis of patients with similar clinical characteristics can be very different. In addition, the analysis of messenger RNA expression, microRNA expression, promoter methylation and DNA copy number in 489 high-grade SOCs from The Cancer Genome Atlas (TCGA) indicated different molecular signatures in tumors with serous histology. Moreover, a defective homologous recombination is critical for choosing a maintenance therapy in the first-line situation at present [6]. Therefore, by using gene signatures as biomarkers, personalized therapy can be optimized and, subsequently, the clinical outcome of advanced-stage SOC patients could be improved.

The tumor microenvironment (TME) consists of a complex system of immune, inflammation and myeloid-derived suppressor cells as well as fibroblasts, macrophages, vasculature and various soluble factors [7,8]. Generally, this dynamic biological system sustains the development of malignant tumors and plays a crucial regulatory role in the emergence of resistance to chemotherapy and targeted therapy [9,10]. Previous studies have shown that upregulated expression of TME genes might correlate with the clinical outcome of OC patients. In particular, increased levels of the serum-soluble tumor necrosis factor receptor 2 may indicate a poor prognosis, as it was significantly higher in samples collected from malignant tumors compared to samples from benign neoplasms [11]. Furthermore, the presence of tumor-associated macrophages was associated with tumor progression, including cell invasion, metastasis, angiogenesis and early recurrence [8,12,13]. In contrast, another analysis of 1731 tumors obtained from SOC patients indicated that high levels of activated dendritic tumor-infiltrating lymphocytes were associated with a good prognosis [14,15].

Taken together, these results indicate that exploring the TME has an important role in assessing the prognosis of OC patients and angiogenesis or immune checkpoint inhibitors, which are currently in phase II/III clinical trials, and represent targeted therapies to TME components [16,17,18]. Considering that the TME in OC tumors is highly heterogeneous [19], a TME-related gene signature might improve the accuracy of predicting clinical response to targeted therapy. Consequently, with this study, we are aiming to design a gene signature based on TME-related genes and, further, we will attempt to use the developed signature to predict the response to immune checkpoint inhibitors.

## 2. Materials and Methods

### 2.1. Data Download and Preprocessing

The Genomic Data Commons Application Programming Interface portal was used to download the gene expression profiles from SOC patients. The data based on TCGA and the cohort contained 379 RNA sequencing samples. We processed the RNA sequencing data of TCGA by removing the samples without clinical follow-up information and retaining the expression profile of TME-related genes only for FIGO stage III and IV samples. Finally, 347 SOC samples were included.

Moreover, for external validation, the Australian OC cohort from the International Cancer Genome Consortium (ICGC) was downloaded. Samples without clinical follow-up information were removed and only FIGO stage III and IV samples were retained. After this process, 93 samples from the ICGC cohort remained.

TME-related genes were obtained from published research [20,21,22,23,24,25,26], which, after sorting, provided 4061 genes (Appendix A).

### 2.2. Identification of Molecular Subtypes Using Non-Negative Matrix Factorization (NMF) Algorithm

The expression levels of the 4061 TME genes were extracted from TCGA expression profile and a univariable Cox analysis was performed using the coxph function [27,28] to obtain prognostic genes of OC (*p* < 0.01). NMF was used to cluster the samples for 50 iterations by extracting biological correlation coefficients and internal feature structures of the prognostic gene expression matrix [29]. The number of clusters k was from 2–10. When k = 3, the cluster demonstrated proper stability and performance, resulting in the clusters 1, 2 and 3 (C1, C2 and C3). The selection was based on cophenetic and residual sum of squares (rss) (Appendix A).

### 2.3. Comparison of Immune Scores between Clusters

Using the Microenvironment Cell Populations (MCP) counter [26], we compared the immune scores among CD8+ T cells, cytotoxic lymphocytes, B cells, neutrophils, monocytic cells, myeloid dendritic cells, endothelial cells and fibroblasts between C1, C2 and C3.

### 2.4. Sample Preparation

The 347 TCGA samples were divided into training and testing cohorts with the caret package, which is a standard tool in building a prediction model [30]. This package focuses on simplifying model training and tuning across a wide variety of modeling techniques. It also includes methods for preprocessing training data, calculating variable importance and model visualizations. To prevent the bias of random allocation from undermining the stability of the subsequent modeling, all samples were put back into random groups 100 times in advance [31]. Herein, the group sampling of training and testing cohorts were performed in a ratio of 7:3. The most suitable training and testing cohorts were selected according to the following conditions: the two cohorts were similar in age distribution, FIGO stage and follow-up time. The gene expression profiles of the two randomly grouped cohorts were similar in the number of classified samples.

### 2.5. Lasso Regression Analysis

We further used lasso Cox regression to reduce the number of genes in the risk model. The lasso method is a regularization estimation used to build a more refined model by constructing a penalty function, thereby compressing some coefficients and setting some coefficients to zero [32]. When applying the glmnet package for lasso Cox analysis, we used 10-fold cross-validation for the model construction and analyzed the confidence intervals (CIs) in each lambda [33].

### 2.6. Construction of Nomogram Combined with Risk Score (RS) and Clinical Features

A nomogram is a method to display the results of the risk model intuitively and effectively, conveniently applied in prediction of the outcome [34]. It uses the length of the scale to represent different variables (RS, clinical features), thereby exhibiting the effect of their values on the outcome. We used TCGA cohort to build a nomogram that combined different prognostic variables.

### 2.7. Prediction Model Evaluation

To compare more than two groups, a Kruskal–Wallis test and one-way ANOVA were used as nonparametric and parametric tests, respectively. The Kaplan–Meier (KM) method was used to calculate the overall survival (OS) and a log-rank test determined the significance. The clinical utility of the prognostic model was evaluated based on the receiver operating characteristic (ROC) curve and decision curve analysis (DCA). We used the concordance index (C-index) to compare the prediction performance between different models. All statistical analyses utilized R software (version 3.6.1).

### 2.8. Immunotherapy Prediction

To predict the response to immunotherapy, we selected the IMvigor 210 cohort with urogenital cancer patients who received a programmed death ligand 1 (PD-L1) blocking therapy [35]. The IMvigor210CoreBiologies package was used to extract clinical information and gene expression profile information, which were converted from counts to transcripts per million and further log2 processed. This cohort consists of 348 samples, including 232 deaths and 116 censored. After PD-L1 treatment, samples were classified into the following categories according to the patient’s response: complete response (CR), partial response (PR), stable disease (SD) and progressive disease (PD). Among them, CR and PR are recognized as patients who respond to immunotherapy. SD and PD are recognized as patients who do not respond to immunotherapy. The value for the neoantigen (NEO) and the tumor mutation burden (TMB) of each sample is also given in this cohort. For further details, please refer to Appendix A.

## 3. Results

### 3.1. Immune Scores Indicate Different Patterns According to Molecular Subtypes in OC

There were 347 OC samples from TCGA and 93 from the ICGC after preprocessing (Table 1). In the expression matrix of the whole TCGA cohort with 4061 genes, 87 prognostic TME-related genes (*p* < 0.01) were obtained by univariable Cox analysis (Appendix A). Out of these genes, three clusters were built using NMF as described in Section 2.2. TCGA-OC samples were than divided into three clusters: C1, C2 and C3 (Figure 1a). Comparison of OS and progression-free survival (PFS) between the three clusters showed significant differences (Figure 1b–g). The majority of the immune scores, including CD8+ T cells, cytotoxic lymphocytes, B cells, neutrophils, monocytic cells, myeloid dendritic cells, endothelial cells and fibroblasts, revealed significant differences among the subtypes as well (Figure 1h–o).

According to TCGA research network, SOC can be divided into four molecular subtypes according to their expression of mRNA: differentiated, immunoreactive, mesenchymal and proliferative [36]. While these subtypes are based on the expression of all genes from TME and OC cells, our three clusters model included exclusively TME-related genes. When comparing C1, C2 and C3 with the existing TCGA classification based on the typing of all genes, we found that the immunoreactive subtype accounted for 55.43% of C1, mesenchymal subtype for 52.04% of C2 and proliferative subtype for 50.98% of C3 (Figure 1p).

### 3.2. Establishment of a Predictive RS Based on TME-Related Genes

The training cohort was composed of 243 samples and the testing cohort of 104 samples (Table 2). The results of the chi-squared test showed no significant differences in FIGO stage, grade, chemotherapy and recurrence status between them (*p* > 0.05), which confirmed the success of randomization.

The univariable Cox proportional hazard regression was conducted on the training cohort using the survival coxph function. Ninety-two prognostic TME-related genes were subsequently identified within the threshold of *p* < 0.01 (Appendix A). The glmnet package [33] was used to further narrow the range of genes, while still maintaining high accuracy. From the change trajectory of independent variables, we can see that as lambda gradually decreased, the number of independent variable coefficients tending towards zero gradually increased (Appendix A). We used 10-fold cross-validation to build the model and to analyze the CI under each lambda. We saw that the model performance was best when lambda was −2.8700 (Appendix A). For this reason, we selected 24 genes according to the minimum lambda = 0.0566 as candidate genes. The Akaike information criterion algorithm was used to reduce the number of genes further and, finally, 11 genes were obtained, constructing this formula:risk score (RS) =−(5.9449 × *SNRPA1*)− (6.9887 × *CCL19*)− (4.4685 × *CXCL11*)− (6.9226 × *CDC5L*)− (6.1777 × *APCDD1*)− (8.9229 × *LPAR2*)+ (0.2541 × *PI3*)+ (1.7480 × *PLEKHF1*)+ (5.4819 × *CCDC80*)+ (0.3243 × *CPXM1*)+ (0.7416 × *CTAG2*).

The p-ROC package was used to analyze the prognostic classification effect of the RS in the training cohort. The 5-year ROC of the model was 0.78. According to the expression of the RS, we were able to divide the samples into a high-risk group (HRG) and a low-risk group (LRG) with the median as our cut-off.

A KM survival analysis showed that the prognosis of the HRG was significantly poorer than that of the LRG (Figure 2a). To validate the accuracy of our model, we applied the same analysis on TCGA testing cohort, the entire TCGA cohort (TCGA training and testing cohort taken together) and the Australian ICGC cohort (Figure 2b–d). All KM curves showed a significant difference between the HRG and the LRG. In the three TCGA cohorts, the 5-year AUCs of the model were 0.78, 0.78 and 0.77. In the ICGC cohort, the 1-year AUC was 0.73. Meanwhile, the 3-year and 5-year AUCs were 0.59, partly because the median follow-up time used by the ICGC was shorter than the one used by TCGA.

### 3.3. RS Assessment in Subgoups Presenting Different Clinical Features or Mutation Statuses

Further, we categorized patient subgroups by age (≤60 years and >60 years), FIGO stage (III, IV), recurrence and chemotherapy status. In addition, we divided the samples into the HRG and LRG based on the already mentioned RS. In most cases, we found a significant difference in the prognosis between the HRG and the LRG (Figure 3a–g). Only the no chemotherapy status showed no significant difference in OS between both groups (Figure 3h). Our data indicate that our prediction model is able to predict OS according to age, FIGO stage III and IV, recurrence and chemotherapy status.

We then extracted the mutation status from all TCGA OC exon sequencing samples. Our aim was to verify whether our model could apply to different mutations as identified by TCGA. More specifically, *TP53*, *TTN*, *MUC16* and *CSMD3*, genes with the highest mutation frequency in OC, were included (Appendix A). For our TCGA cohort, we identified mutation status in 250 samples (in 97 samples the mutation status was unclear). All subgroups confirmed significant differences in the OS between the HRG and the LRG (Figure 3i,k,l,n–p). The subgroup without a *TP53* mutation and the one with a *MUC16* mutation showed no significance (Figure 3j,m).

### 3.4. TME-Related Genes Correlate with Clinical Outcome

We next employed univariable and multivariable Cox regression to analyze the relationship between RS and OS (Table 3). Our analysis revealed that RS is an independent risk factor with a hazard ratio (HR) of 2.499. A similar result was indicated by residual disease (HR = 1.70). In contrast, chemotherapy is an independent protective factor (HR = 0.35). FIGO stage, grade and recurrence showed no significance.

Consequently, we constructed a nomogram model including residual disease, RS and chemotherapy status. It showed that RS had the greatest impact on the prediction of survival rate (Figure 4a). Calibration plots visualized the 3-year and 5-year performance and showed that our nomogram is close to the 45° line that represents the optimum in prediction [37] (Figure 4b). In the 3-year ROC curve, the AUC of the nomogram was larger than other clinical variables, which can be seen in Figure 4c. However, in the 5-year ROC curve, the RS exceeded the nomogram and had the largest AUC (Figure 4d). Further, we used DCA to evaluate the effectiveness of the model [38]. The standardized net benefit confirmed the 3-year predictive value in OS of the nomogram (Figure 4e) and the 5-year predictive value in OS of the RS (Figure 4f) when compared using a single clinical variable.

### 3.5. The 11-Gene Signature Risk Model as a Novel Predictive RS in OC

To compare the prediction performance of our 11-gene signature with other models, we selected four other reported risk models: a 5-gene [39], an 8-gene [40], a 10-gene [41] and a 7-gene signature [42]. To make them comparable, we calculated the RS of each dataset using the same method (multivariable Cox regression analysis). In the validation of these results, we included the corresponding genes in these four models and then evaluated the ROC. The samples were subsequently divided into the HRG and the LRG according to the median RS value. The prognosis for the HRG and LRG was significant across all four models. However, the ROC curves showed a lower AUC, therefore, they were poorer in predicting the prognoses when compared to our model (Figure 5a–d).

The restricted mean survival (RMS) package was used to calculate the C-index of all prognostic signatures. Our model had the highest C-index with 0.641 (Figure 5e). With the RMS time, an evaluation of the predictive effect of the signatures at different time points was possible. As a result, our gene signature performed best at a time period greater than 60 months. This indicates that our model is not only suitable for predicting patient survival greater than five years, but is also the leading predictor at less than 60 months when compared with the other models (Figure 5f).

### 3.6. The 11-Gene Signature Risk Model Validation in Another Gynecological Cancer

We ran gene set enrichment analysis on samples within the HRG and LRG in order to show a possible over-proportional representation of essential signaling cascades. Thereby, we found the following immune and metabolism pathways with the associated normalized enrichment score (NES) and the adjusted *p*-value (*q*-value) enriched in the HRG (Figure 6a):*KEGG_PATHWAYS_IN_CANCER*(NES = 0.4600, *q* = 0.0049),*KEGG_T_CELL_RECEPTOR_SIGNALING_PATHWAY*(NES = 0.5604, *q* = 0.0030),*REACTOME_ADAPTIVE_IMMUNE_SYSTEM*(NES = 0.4685, *q* = 0.0010),*REACTOME_CYTOKINE_SIGNALING_IN_IMMUNE_SYSTEM*(NES = 0.4100, *q* = 0.0030),*REACTOME_METABOLISM_OF_LIPIDS*(NES = 0.4862, *q* = 0.0010).

Our analysis showed differences in the RS among OC subtypes as identified by TCGA [36]. More specifically, the RS was significantly higher in the mesenchymal subtype than in the differentiated, immunoreactive and proliferative subtypes. Moreover, the RS in the immunoreactive subtype was significantly lower than in the differentiated subtype. In conclusion, to a certain extent, our model is able to distinguish between the different molecular OC subtypes (Figure 6b).

Along with the pathway analysis in Figure 6a, we extracted the single gene expression of representative genes for generic pathway targets in gynecological oncology: immune checkpoints, DNA replication, mismatch repair and epithelial–mesenchymal transition (EMT). We then analyzed the correlation between the RS and these genes. RS was negatively correlated with immune checkpoint genes (*PDCD1*, *CD274* and *CTLA4*). In contrast, they were positively correlated with EMT-related genes (*FAP*, *TAGLN* and *LOXL2*) (Figure 6c).

To validate our 11-gene risk model in another gynecological cancer, we selected the cervical squamous cell carcinoma (CESC) cohort from TCGA. In this aim, our 11-gene RS was able to divide CESC samples into the HRG and LRG. The KM curve indicated that the prognosis of patients included in the HRG was significantly worse (*p* = 0.024) (Figure 6d).

Moreover, studies have shown that through the integration of genomic and proteomic characteristics, three molecular subtypes of cervical cancer can be distinguished. These are classified as the following: hormone, PI3K-AKT and EMT [43]. Given the predictive value of RS in OC molecular subtypes, the RS was able to differentiate among CESC subtypes: for the EMT subtype, the RS was significantly higher than the one obtained for the hormone subtype (Figure 6e).

We used the MCP counter and Estimation of Stromal and Immune cells in Malignant Tumor tissues using Expression data (ESTIMATE) to calculate different immune (cell) scores of the CESC samples and then calculated their correlation with the RS. It showed that the RS was negatively correlated with the stromal, immune and ESTIMATE score as well as with many immune and stromal cell lines (Figure 6f). 

### 3.7. Prediction of Response to Immunotherapy Based on 11-Gene Risk Model

To date, immunotherapy does not play a major role in the therapy of OC [44,45]. However, new predictive biomarkers would be crucial for patient selection for immunotherapy. To explore whether our 11-gene model and subsequently the RS could predict the response to immunotherapy, we analyzed our immune signature in patients with urogenital cancer who received immunotherapy. We identified the IMvigor 210 cohort [35], which included gene expression and clinical responses from patients treated with PD-L1 blocking therapy. Comparing the HRG and LRG which were previously divided according to the RS, complete and partial immune responders accounted for 17.3% in the HRG and 27.3% in the LRG. Taken together, our results confirm the accuracy of the RS in another cancer entity (Figure 7a).

We combined our RS, the NEO and the TMB from the IMvigor 210 cohort [35] by logical regression to determine the treatment response and found that the AUC of this combination was as high as 91.7%. This is higher than both of the other single values: NEO (AUC = 0.701) and TMB (AUC = 0.636) (Figure 7b). We calculated the correlation between RS, NEO and TMB, as well as different immune scores using the MCP counter and found that the RS negatively correlated with all of them except fibroblasts (Figure 7c). Moreover, the RS showed significantly higher values (*p* = 0.035) in patients with PD compared to patients with CR. However, it was not able (*p* > 0.05) to distinguish between PR and SD (Figure 7d). Applying our RS to the IMvigor 210 cohort, we were able to significantly distinguish an immune cell score (IC score) of 0 from 1 (*p* = 0.008) as well as from 2+ (*p* < 0.001). The IC score categorizes tumors according to the percentage of PD-L1-positive immune cells (macrophages, dendritic cells and T cells): IC0 = 0%, IC1 < 5%, IC2+ ≥ 5%. In general, there was a significantly negative correlation between the RS and the IC score (Figure 7e). Additionally, an immunophenotypic analysis showed a low RS to be significantly associated with an immune inflamed phenotype (data not shown).

## 4. Discussion

The main finding of our study is the establishment of a powerful predictive model based on a TME-related gene signature in advanced-stage SOC.

As discussed in more detail below, we think it is crucial to explore prognostic markers given the TME’s heterogeneity [46,47,48], to predict clinical response to targeted therapies.

In our study, 347 advanced-stage SOC samples and 4061 TME-related genes from TCGA were analyzed. We applied biostatistical methods to establish three molecular clusters. Additionally, immune scores, representing the proportion of several TME-related cell types, were compared between the clusters and showed significant differences. These results were in line with the concept of a heterogeneous TME. Furthermore, correlation analyses with clinical parameters such as OS and PFS resulted in significant differences between these clusters as well.

In the present study, we showed that our 11-gene risk model (*SNRPA1*, *CCL19*, *CXCL11*, *CDC5L*, *APCDD1*, *LPAR2*, *PI3*, *PLEKHF1*, *CCDC80*, *CPXM1* and *CTAG2*) is able to indicate the prognosis of advanced-stage SOC patients. Previous studies investigated several genes, included in our signature, in an oncological context. For example, the mTOR pathway is able to participate in the malignant progression of hepatocellular carcinomas by activating *SNRPA1* [49]. In contrast, an elevated expression of *SNRPA1* would decrease our RS in OC patients. However, the role of *CTAG2*, coding for one of several tumor-associated autoantibodies that may have the ability to allow earlier diagnosis of SOC [50], is in line with our results. Even *CCL19*, transcribed by fibroblasts, inhibits the growth of lung cancer by promoting local anti-tumor T cell response [51] and *CXCL11* is known to be used as an enhancer of vaccine-induced CD8+ T cellular immunity [52].

The multivariable Cox regression analysis showed that the RS was an independent prognostic risk factor and remained robust when verified with internal and external cohorts. Next, we compared the HRG and LRG in defined subgroups, such as with or without a *TP53* mutation. While low-grade SOC is frequently characterized by *KRAS*, *BRAF*, *PTEN* and *CTNNB1* mutations, high-grade SOC expresses *TP53* mutations as its main molecular feature [53]. However, the correlation between *TP53*, the type of mutation and the prognosis of SOC is still controversial [54,55,56,57]. While our study is not able to close this gap in the knowledge, the RS showed a highly significant difference between the HRG and the LRG in the *TP53* mutated samples. Consequently, the RS can offer a resilient risk stratification in the huge group of high-grade SOC patients.

Nomograms are well established biostatistical methods in predicting the prognosis of patients with OC [58,59,60,61,62]. However, for the first time, we have constructed an advanced-stage SOC nomogram based on the TME-RS. ROC curve and DCA confirmed that the RS can accurately evaluate the prognosis of patients. When we compared our risk model to four other prognostic risk signatures from Wang et al. [39], Yue et al. [40], Wang et al. [41] and Sabatier et al. [42], our C-index demonstrated the highest AUC. These results indicate that the overall performance of our proposed model is superior to others. When we transferred this RS model to another gyneco-oncological entity, cervical cancer, it confirmed the ability to safely differentiate between the prognoses of an HRG and an LRG and between several molecular subtypes, all of which demonstrate potential pan-cancer usability.

Furthermore, our RS was strongly correlated with EMT-related genes. In contrast, immune checkpoints, DNA replication and mismatch repair were negatively correlated to the RS. Theoretically, a higher degree of immune infiltration in the LRG, partly represented through immune checkpoints, could account for greater immune defense capacity and consequently a better prognosis. Vice versa, the positively correlated EMT genes might cause OC’s higher tendency to metastasize and a worse prognosis in the HRG.

In recent years, several clinical trials investigating immune checkpoint inhibitors in gynecological cancers were initiated and have attracted considerable attention [63]. Nevertheless, recently, study results on the PD-L1 inhibitors avelumab [44] and atezolizumab [45] revealed disappointing results in OC, as neither reached their primary endpoint of PFS. Additionally, for the programmed death 1 inhibitor pembrolizumab, we are still lacking encouraging phase III studies. This might be caused by the relatively small TMB in OC compared with other entities such as endometrial cancer [64]. Since immunotherapy gene datasets with clinical correlation for OC are not available, we selected another dataset which was obtained from samples of metastatic urothelial cancer treated with a PD-L1 blockade [35]. Hereby, our RS confirmed its performance in another cancer entity. Moreover, the accuracy of the prediction of a response to an anti-PD-L1 therapy is immensely increased by up to 91.7% if the RS is combined with NEO and TMB. This power is supported by the RS’s ability to significantly differentiate CR from PD and distinguish between IC0 and IC1 as well as IC2+. Interestingly, our RS is negatively correlated with the PD-L1 expression, which could be caused by a known phenomenon: “adaptive immune resistance” [65]. Here, when tumor cells are attacked by the immune system, interferon gamma is overexpressed through the active anti-tumor immune response. This induces the expression of PD-L1, indicating a strong attack by the immune system and in our case possibly leading to a better prognosis of the LRG patients [66,67]. The prognostic ability of our RS is supported through the correlation of a low RS with an immune inflamed phenotype, one of the three different phenotypes of immunology that most solid tumors exhibit [68]. Checkpoint inhibitors have already exerted anti-tumoral effects in this immunophenotype [69], which confirms our findings.

Of course, there are some limitations to this study. Firstly, as this is a retrospective study and interpatient variability is wide, and the results should be further verified in prospective studies. Secondly, all the SOC transcriptome data used were derived only from TCGA and ICGC databases. Lastly, estimations on the possible causality between gene expression and clinical findings, drawn in the discussion, need to be validated via further translational research trials. Hence, this unselected view on all TME-related genes was one of the initial aims of this study and can also be seen as a potential strength.

## 5. Conclusions

In conclusion, our study proposes a TME-related risk model to be implemented in the assessment of advanced-stage SOC patients. First, the accuracy of our RS to predict the biological response of cancer samples to an anti-PD-L1 immunotherapy could be clinically important. Even though several clinical trials failed to show a clinical benefit for checkpoint inhibition in OC patients [44,45], our RS might improve patient selection in future trials. Lastly, the establishment of a novel prognostic tool to predict a patient’s prognosis based on an 11-gene signature might elevate prognostic accuracy, therefore encouraging users to combine several gene signatures. In this aim, the combination of this TME-related risk model with the gene signature based on lipid metabolism [70], which was recently published by our group, might be an option.

## Figures and Tables

**Figure 1 cancers-13-03343-f001:**
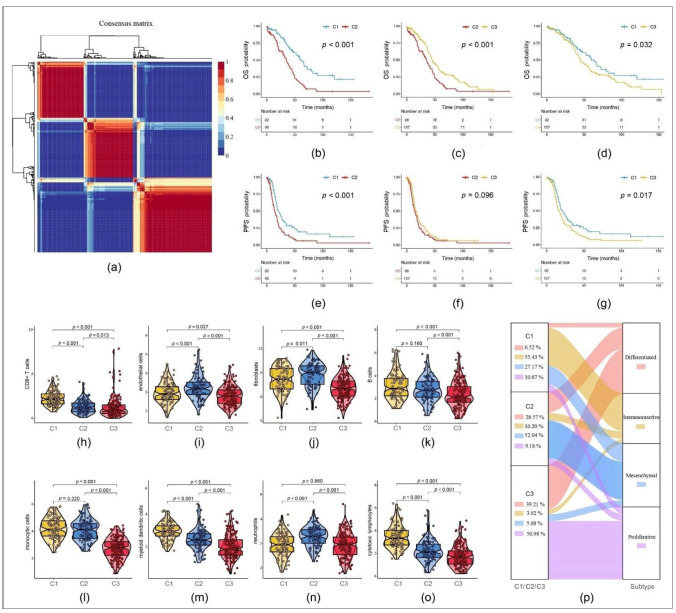
Comparison of the three clusters 1, 2 and 3 (C1, C2 and C3): (**a**) Consensus map clustered via the non-negative matrix factorization (NMF) algorithm. (**b**–**g**) Overall survival (OS) and progression-free survival (PFS) showed significant differences. (**h**–**o**) Immune scores of cells of the tumor microenvironment (TME) showed significant differences. (**p**) Percentage of the four molecular subtypes accounting for each of the three clusters.

**Figure 2 cancers-13-03343-f002:**
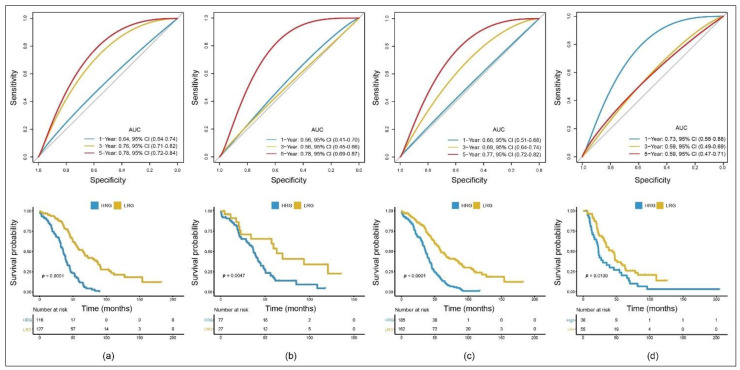
Construction and validation of the TME-related 11-gene risk score (RS) for serous ovarian cancer (SOC) with 1-, 3- and 5-year receiver operating characteristic (ROC) and OS Kaplan–Meier (KM) curves within the different cohorts: (**a**) TCGA training cohort; (**b**) TCGA testing cohort; (**c**) entire TCGA cohort; (**d**) Australian ICGC cohort.

**Figure 3 cancers-13-03343-f003:**
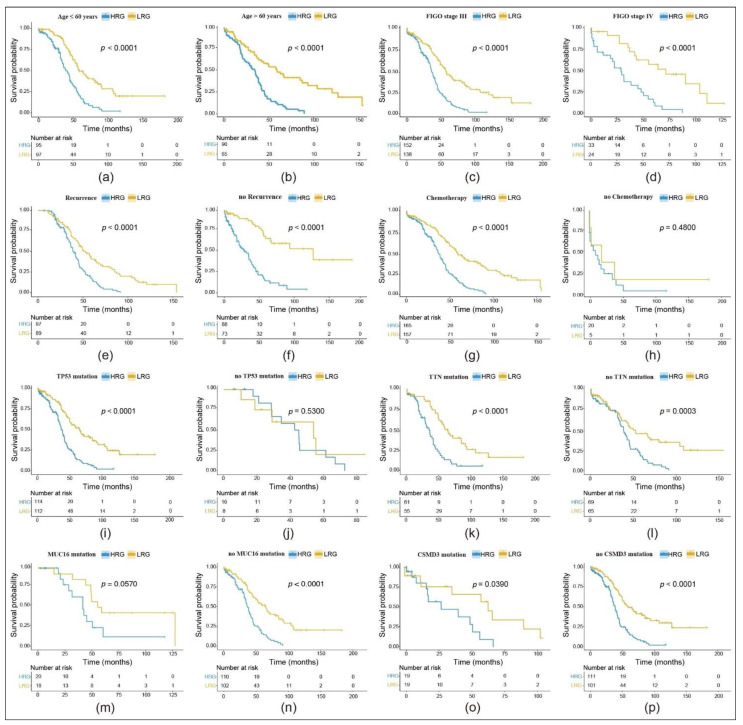
Evaluation of the 11-gene risk model in different subgroups with OS KM curves comparing the high-risk group (HRG) and the low-risk group (LRG): (**a**–**h**) Subgroups with different clinical features; (**i**–**p**) subgroups with different mutations.

**Figure 4 cancers-13-03343-f004:**
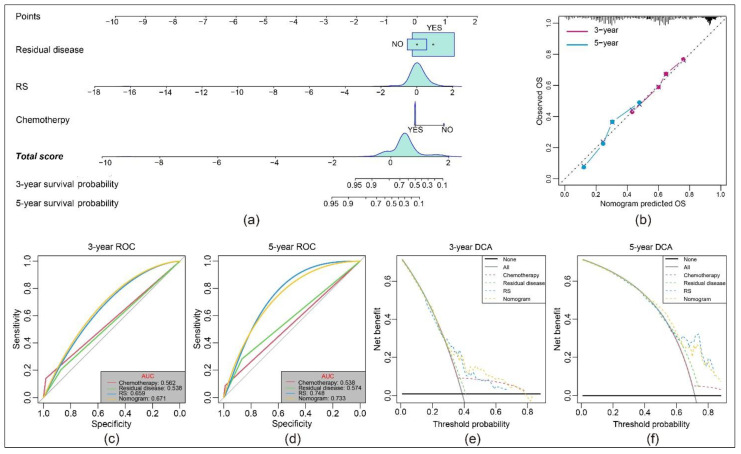
Clinical value of the predictive model: (**a**) Nomogram predicting the 3- and 5-year OS for patients. The points identified on the point scale of each variable are totaled. Finally, beneath the total points, the probability of 3- or 5-year survival is projected on the scales below. (**b**) Calibration curves for the nomogram predicted 3- and 5-year OS for patients in relation to actual survival. (**c**,**d**) ROC curves of the nomograms compared with other clinical variables. (**e**,**f**) Decision curve analysis (DCA) curves can evaluate the clinical benefit of the nomograms and their potential scope of application. Black indicates that all samples are negative and none are treated, therefore the net benefit is zero. Gray indicates that all samples are positive and all are treated. The x-axis represents the threshold probabilities.

**Figure 5 cancers-13-03343-f005:**
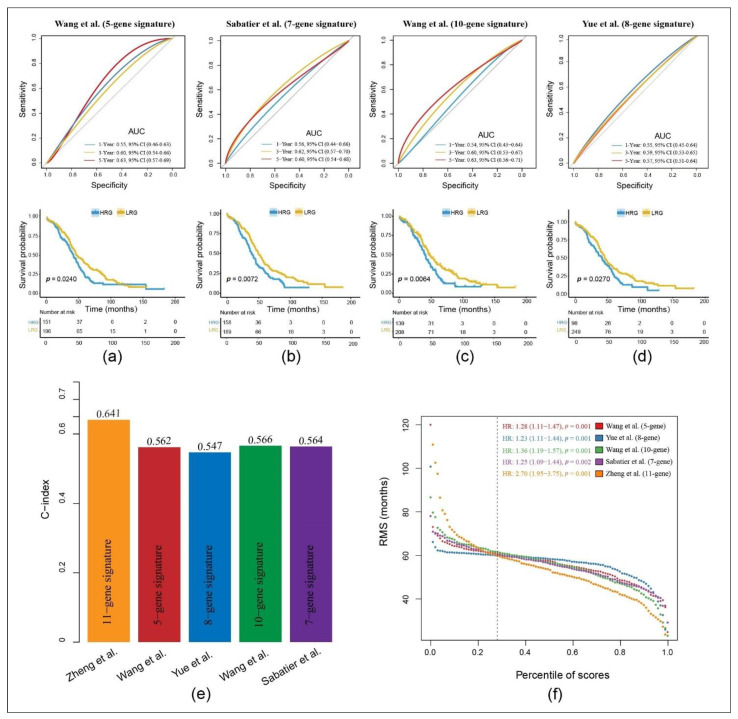
Comparison of the 11-gene risk model with other established models: (**a**–**d**) ROC and KM curves of four other published gene-signatures. (**e**) Concordance index (C-index) of the five prognostic risk models including our model, which has the highest C-index. (**f**) Restricted mean survival (RMS) time curve of all five prognostic risk models, revealing an overlap of 60 months.

**Figure 6 cancers-13-03343-f006:**
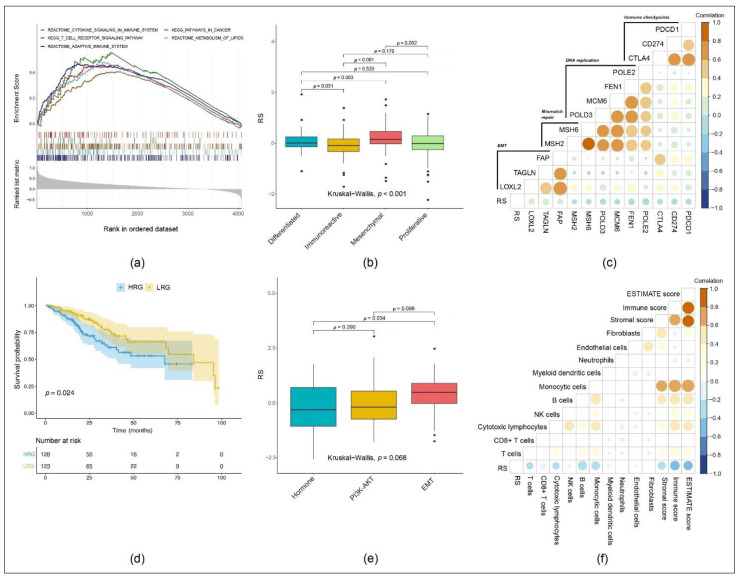
Functional analysis and pan-cancer verification of the model: (**a**) The main pathway groups overexpressed in the HRG. (**b**) RS comparison of the four molecular subtypes of TCGA OC. (**c**) Correlation between RS of OC samples and expression of representative genes for generic pathway targets in gynecological oncology (Pearson method). (**d**) OS of the HRG and the LRG in the KM curve when applying the 11-gene signature on the cervical squamous cell carcinoma (CESC) cohort. (**e**) RS comparison of the three molecular subtypes of TCGA CESC. (**f**) Correlation between RS of CESC samples and immune scores (Pearson method).

**Figure 7 cancers-13-03343-f007:**
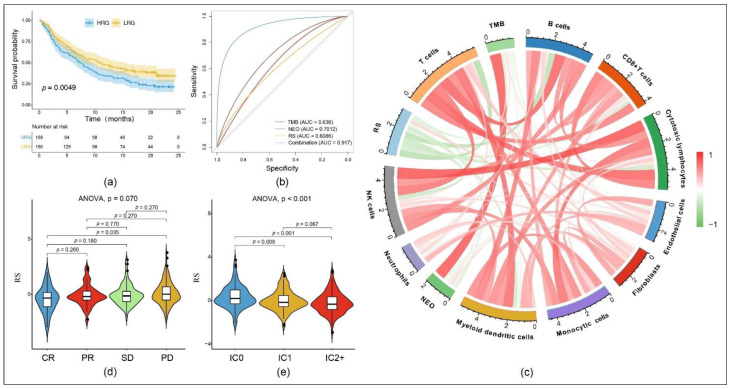
Prediction of potential immunotherapy using the RS in the IMvigor 210 cohort with human metastatic urothelial carcinoma samples from patients receiving programmed death ligand 1 (PD-L1) blocking immunotherapy: (**a**) The KM curve dividing the cohort into an HRG and an LRG according to the RS. (**b**) ROC curve for the prediction of treatment response to PD-L1 blocking therapy comparing tumor mutation burden (TMB), neoantigen (NEO), RS and the combination of all three. (**c**) Correlation between RS, immune cell score (IC-score), TMB and NEO. (**d**) RS according to the effectiveness of immunotherapy expressed as complete response (CR), partial response (PR), stable disease (SD) or progressive disease (PD). (**e**) RS according to different IC scores.

**Table 1 cancers-13-03343-t001:** Sample information from The Cancer Genome Atlas (TCGA) and International Cancer Genome Consortium (ICGC) Australian ovarian cancer (OC) cohort.

Clinical Feature		TCGA	ICGC-Australian
Event	Censored	125	19
	Dead	222	74
FIGO stage	III	290	79
	IV	57	14
Grade	G1	1	
	G2	35	
	G3	302	
	G4	1	
	None	8	
Chemotherapy	Yes	322	
	No	25	
Recurrence	Yes	186	
	No	161	
Age	≤60	192	
	>60	155	
Residual disease	No	55	
	1–10 mm	25	
	11–20 mm	164	
	>20 mm	69	
	Unknown	34	

**Table 2 cancers-13-03343-t002:** Comparison of TCGA training and testing cohort.

Clinical Feature		TCGA Training Cohort	TCGA Testing Cohort	*p*-Value
Event	Censored	92	33	0.3333
	Dead	151	71	
FIGO stage	III	201	89	0.6165
	IV	42	15	
Grade	G1	1	0	0.5779
	G2	22	13	
	G3	212	90	
	G4	1	0	
	None	7	1	
Chemotherapy	Yes	227	95	0.6481
	No	16	9	
Recurrence	Yes	128	58	0.6803
	No	115	46	
Age	≤60	140	52	0.1913
	>60	103	52	
Residual disease	No	41	14	0.8823
	1–10 mm	114	50	
	11–20 mm	17	8	
	>20 mm	46	23	
	Unknown	25	9	

**Table 3 cancers-13-03343-t003:** Univariable and multivariable Cox regression to analyze the relationship between the RS and clinical prognosis.

Variables	Univariable Analysis			Multivariable Analysis		
	HR	95% CI	*p*-Value	HR	95% CI	*p*-Value
FIGO stage	1.16	0.82–1.66	0.408	1.24	0.86–1.78	0.251
Grade	1.03	0.67–1.58	0.884	1.12	0.73–1.71	0.620
Chemotherapy	0.29	0.18–0.48	<0.001	0.35	0.20–0.59	<0.001
Recurrence	1.15	0.86–1.54	0.353	1.21	0.89–1.65	0.227
Age	0.79	0.60–1.04	0.097	0.79	0.59–1.05	0.101
Residual disease	2.06	1.29–3.26	0.001	1.70	1.06–2.73	0.027
RS	2.19	1.53–3.15	<0.001	1.73	1.17–2.55	<0.006

## Data Availability

Publicly available datasets were analyzed in this study and can be found here: https://www.cancer.gov/tcga and https://daco.icgc.org/ (accessed on 1 April 2021). Data generated by the authors are shown in this paper or in the Appendix A. Further data are available on request from the corresponding author, if they are not shown somewhere else.

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
