# Peer review of "Identification of a Novel Tumor Microenvironment Prognostic Signature for Advanced-Stage Serous Ovarian Cancer"

_cancers, 2021, doi:10.3390/cancers13133343_

Round 1
Reviewer 1 Report
The publication titled: “Identification of a Novel Tumor Microenvironment Prognostic 2 Signature for Advanced-Stage Serous Ovarian Cancer” is a very well written. It is looking to establish gene set expressed in tumor microenvironment that correlates with clinical outcomes. The authors further validated their score by using additional ovarian ca samples from TCGA, as well as gene sets from other cancers. Interestingly, the authors were able to use their score to correlate with responses in urogenital cancers treated with immune-chemotherapy in retrospective-prospective fashion.
Ability to identify high risk groups in low clinical risk group and low risk group in high clinical risk groups certainly is important to better match therapy recommendations. The correlation of RS with immune therapy responses is not surprising as the observation of immune activation and prognosis correlation has been shown across multiple cancers. The ability of combination of RS, NEO and TMB to correspond with responses is an interesting finding although the usefulness in clinic would have to undergo other validation steps.
Probably the biggest limitation is that throughout authors use bulk RNA seq data therefore contributions from specific cells is unknown. Although they chose >4000 genes that have been published as relevant in tumor microenvironment – the tumors in general and ovarian ca specifically have also been found to have varied levels of these genes.
The methods do not adequately describe selection of TCGA cohorts (training and validation).
methods do not address the number/completeness of samples from the prospective immune treatment study and no details of the treatment regimen/selection of patients are provided which are relevant to the outcomes.
methods should provide additional detail of the analysis
Author Response
Reviewer 1:
The publication titled: “Identification of a Novel Tumor Microenvironment Prognostic Signature for Advanced-Stage Serous Ovarian Cancer” is a very well written. It is looking to establish gene set expressed in tumor microenvironment that correlates with clinical outcomes. The authors further validated their score by using additional ovarian ca samples from TCGA, as well as gene sets from other cancers. Interestingly, the authors were able to use their score to correlate with responses in urogenital cancers treated with immune-chemotherapy in retrospective-prospective fashion.
Ability to identify high risk groups in low clinical risk group and low risk group in high clinical risk groups certainly is important to better match therapy recommendations. The correlation of RS with immune therapy responses is not surprising as the observation of immune activation and prognosis correlation has been shown across multiple cancers. The ability of combination of RS, NEO and TMB to correspond with responses is an interesting finding although the usefulness in clinic would have to undergo other validation steps.
Probably the biggest limitation is that throughout authors use bulk RNA seq data therefore contributions from specific cells is unknown. Although they chose >4000 genes that have been published as relevant in tumor microenvironment – the tumors in general and ovarian ca specifically have also been found to have varied levels of these genes.
Thank you for your valuable comment. In this study, a large amount of RNA-seq data and chip cohort data were used for a retrospective research. Like many previous transcriptome studies based on tumor tissue level, it is difficult to perform quantitative analysis at the cellular level on a large scale. In fact, tumor heterogeneity leads to significantly different tumor microenvironments in different patients. Specific cell research would be the most effective method, however it is very difficult to achieve in this dimension. Therefore, we used the bulk RNA seq data.
The methods do not adequately describe selection of TCGA cohorts (training and validation).
As for TCGA's choice of training and validation cohort, caret package was used to divide 347 TCGA samples into training and testing groups. We added further informations on this package in the manuscript.
Methods do not address the number/ completeness of samples from the prospective immune treatment study and no details of the treatment regimen/ selection of patients are provided which are relevant to the outcomes. Methods should provide additional detail of the analysis.
We added paragraph 2.8. and Table S2 to the manuscript to answer this issue.
Reviewer 2 Report
The authors (Zheng et al.) have developed a novel prognostic tumor microenvironment related 11 gene signature for advanced-stage serous ovarian cancer based on TCGA and ICGC cohorts using gene expression data and regularized Cox regression on overall survival.
The risk model based on 11 genes showed not only better performance than four previous reported gene signatures in ovarian cancer but was also prognostic in cervical squamous cell carcinoma cohort (CESC) from TCGA and – to a limited extend – predictive for PD-L1 treatment response in the IMVIGOR 210 cohort including patients with locally advanced or metastatic urothelial bladder cancer.
This is well written and structured manuscript with very sound (e.g. careful separation of training and test data) and clinically relevant (e.g. nomogram or decision curve analysis) analysis. Do not have major concerns of the manuscript in its present form and have a few suggestions:
Major issues
- The authors should compare survival outcome of the 11 gene signature risk score with the residual disease after debulking surgery as it is one of the most prognostic factors or include both variables in the multivariable model. If there are too many missing information of that variable in the TCGA cohort would suggest do include analyses of another cohort.
- The authors should explain why AUC for 1yr/3yr/5yr ROC for risk score (RS) is different in Figure 4 compared to Figure 2 and why RS in decision curve analysis is very close to the curve of considering all samples as positive and all are treated. This would indicate that the combined model (nomogram) show good performance and based solely on the risk-score show rather limited perfomance?
- The information on how the data of the IMvigor210 cohort including patients treated with PD-L1 (Rosenberg et al. Lancet 2016) were assessed is missing. The authors should provide this information (for instance how the neoantigen load was derived?)
Minor
- Figure 6 c,f correlation between identical variables should be removed
- Figure 6 a, for GSEA normalized enrichment score and q-values should be provided
- line 352 as "accuracy" is exactely defined ((TP+TN)/N) in this context should not be used loosely instead of AUC,
- line 406 "accuracy" also loosely used instead of performance.
- line 135 lasso is known rather as “regularization” than “compression”
- line 238 “multivariable” should be used instead of “multivariate”
- line 341 “pan-cancer analysis” is not supported by the presented data
Author Response
Reviewer 2:
The authors (Zheng et al.) have developed a novel prognostic tumor microenvironment related 11 gene signature for advanced-stage serous ovarian cancer based on TCGA and ICGC cohorts using gene expression data and regularized Cox regression on overall survival.
The risk model based on 11 genes showed not only better performance than four previous reported gene signatures in ovarian cancer but was also prognostic in cervical squamous cell carcinoma cohort (CESC) from TCGA and – to a limited extend – predictive for PD-L1 treatment response in the IMVIGOR 210 cohort including patients with locally advanced or metastatic urothelial bladder cancer.
This is well written and structured manuscript with very sound (e.g. careful separation of training and test data) and clinically relevant (e.g. nomogram or decision curve analysis) analysis. Do not have major concerns of the manuscript in its present form and have a few suggestions:
Major issues:
- The authors should compare survival outcome of the 11 gene signature risk score with the residual disease after debulking surgery as it is one of the most prognostic factors or include both variables in the multivariable model. If there are too many missing information of that variable in the TCGA cohort would suggest do include analyses of another cohort.
First of all, thank you for the appreciative comments. We included the residual disease in tables 1-3 and further used no macroscopic disease as a reference in the multivariate analysis. The results showed that residual disease is an independent risk factor for the prognosis of patients with advanced-stage serous ovarian cancer (HR=1.70, 1.06-2.73, P value = 0.027). Consequently we modified Figure 4, its legend and the attached paragraph.
- The authors should explain why AUC for 1yr/3yr/5yr ROC for risk score (RS) is different in Figure 4 compared to Figure 2 and why RS in decision curve analysis is very close to the curve of considering all samples as positive and all are treated. This would indicate that the combined model (nomogram) show good performance and based solely on the risk-score show rather limited perfomance?
The reason for the difference in the ROC curve is, that all 347 samples are used to evaluate the stability of the modeling in Figure 2. Whereas in Figure 4, we first performed multivariate cox analysis and then removed the NA ones in all variables. Therefore, the number of samples finally included in the analysis is only 307. Although the final analysis ROC is different from Figure 2, the difference is still small (3 year: 0.69 vs 0.66; 5 year: 0.77 vs 0.75).
After we included the residual disease variables, the DCA analysis was performed again and the results showed the net benefit confirmed the 3-year predictive value in OS of the nomogram (Figure 4e) when compared with using a single clinical variable. But in 5 years, RS has the highest net benefit (Figure 4f). This showed that the combined nomogram has a good performance in the 3-year prediction, while the RS solely performs best in 5-year prediction.
The information on how the data of the IMvigor210 cohort including patients treated with PD-L1 (Rosenberg et al. Lancet 2016) were assessed is missing. The authors should provide this information (for instance how the neoantigen load was derived?).
We added paragraph 2.8. and Table S2 to the manuscript to answer this issue.
Minor issues:
- Figure 6 c, f correlation between identical variables should be removed.
We removed this correlation.
- Figure 6a, for GSEA normalized enrichment score and q-values should be provided.
We added these information in paragraph 3.6..
- Line 352 as "accuracy" is exactely defined ((TP+TN)/N) in this context should not be used loosely instead of AUC.
We changed this expression to AUC.
- Line 406 "accuracy" also loosely used instead of performance.
We changed this expression to AUC.
- Line 135 lasso is known rather as “regularization” than “compression”.
We changed both words with eachother.
- Line 238 “multivariable” should be used instead of “multivariate”.
We changed this in all localizations in the text.
- Line 341 “pan-cancer analysis” is not supported by the presented data.
We removed this expression in this sentence.